# VGPO: Fine-Tuning Speech Autoregressive Diffusion Models with Value Guided Policy Optimization

## Abstract

Autoregressive diffusion models (ARDMs), which generate continuous latent sequences, have recently achieved state-of-the-art zero-shot text-to-speech (TTS) performance. However, fine-tuning these models with reinforcement learning (RL) to directly optimize user-defined reward functions remains an open challenge. In this work, we propose Value-Guided Policy Optimization (VGPO), an actor-critic RL algorithm tailored to ARDMs. We train a causal value model to predict expected future rewards and update the ARDM using gradients from this value model. To validate VGPO, we fine-tune the recently introduced DiTAR model and evaluate it on two tasks: improving F0 variance to enhance expressiveness; and optimizing text log-probability to improve the model's robustness to challenging long text. VGPO can achieve significant improvement in zero-shot TTS expressiveness and robustness, while maintaining naturalness and speaker similarity.

## 1 Introduction

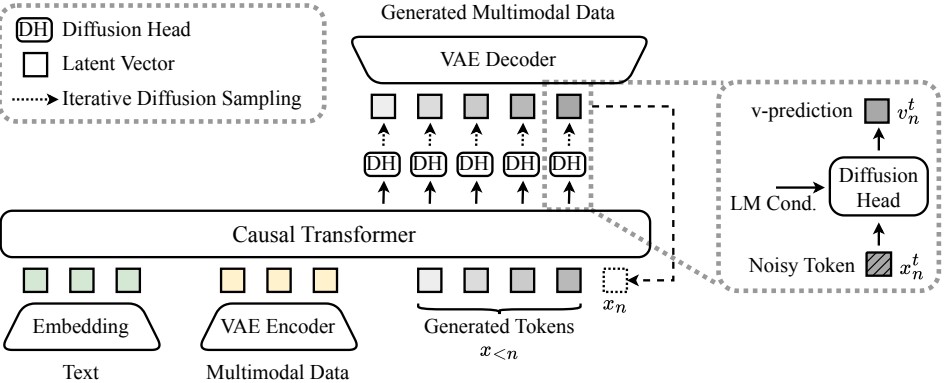

Figure 1: Illustration of a widely adopted autoregressive diffusion model architecture. It is used in state-of-the-art models for speech (Sun et al., 2024; Jia et al., 2025), and image generation (Kou et al., 2025; Sun et al., 2024; NextStep Team et al., 2025). A causal transformer encodes the generation conditions and previously generated tokens, while a diffusion head predicts the next token $x_n$ via diffusion modeling. This architecture is highly efficient, as only the diffusion head is repeatedly evaluated when generating each next token.

Autoregressive diffusion models (ARDMs) represent continuous modalities using latent vectors (continuous tokens) and generate data sequentially by predicting the next token through diffusion modeling. This approach has been adopted in an increasing number multimodal generation models, including audio (Liu et al., 2024; Jia et al., 2025; Yang et al., 2025), image (Li et al., 2024a; Hu et al., 2024), and video generation (Yin et al., 2025; Deng et al., 2024). Unlike the approach of tokenizing data into discrete symbols for next-token prediction, ARDMs offer two key advantages with next-token diffusion (Sun et al., 2024): they preserve fine-grained details while avoiding prohibitively

long sequences, thanks to the compactness of latent continuous representations. This allows for more precise control over the generated content, yielding superior performance on tasks that require high-fidelity details. In particular, several recent works have introduced ARDMs for speech generation (Jia et al., 2025; Sun et al., 2024), achieving state-of-the-art performance in zero-shot text-to-speech.

Reinforcement fine-tuning (RFT) (Ouyang et al., 2022; Rafailov et al., 2023; Xu et al., 2023) is a key stage in the post-training of multimodal generative models. It improves model performance by directly optimizing the expected reward of generated samples under a user-specified reward function $r(\mathbf{x})$. Given a pre-trained reference model $\mu(\mathbf{x})$, RFT optimizes a policy $\rho(\mathbf{x})$ to maximize the expected reward while remaining close to the reference via a divergence regularizer $\mathrm{d}(\cdot\|\cdot)$, which helps mitigate reward hacking (Weng, 2024):

$$\max_{\rho} \mathbb{E}_{\mathbf{x}\sim\rho(\mathbf{x})}[r(\mathbf{x})] - \mathrm{d}(\rho\|\mu) \tag{1}$$

For speech generation, prior work has instantiated $r(\mathbf{x})$ with a variety of reward models, including predicted mean opinion score (MOS) (Chen et al., 2025; 2024; Tian et al., 2025), speaker similarity from speaker encoders (Du et al., 2025; Li et al., 2025), emotion classification accuracy (Anastassiou et al., 2024), and text accuracy measured by automatic speech recognition (ASR) (Zhang et al., 2025). RFT with such rewards has been shown to improve intelligibility, naturalness, speaker identity preservation, and controllability of speech generative models.

Although recent work has extensively explored the network architectures and training techniques of ARDMs, relatively little research has investigated RFT algorithms for these models (NextStep Team et al., 2025; Liu et al., 2025b), leaving this area largely unexplored. In this work, we introduce **V**alue-**G**uided **P**olicy **O**ptimization (VGPO), a novel actor–critic RFT algorithm tailored for ARDMs.

The core component of VGPO is a learned (soft) value function, initialized from a pre-trained ARDM. Given the previously generated tokens and the current partially denoised token, it predicts the expected final reward. Extending prior results in maximum-entropy (MaxEnt) RL (Levine, 2018) and exact energy guidance (Lu et al., 2023) to ARDMs, we prove that the optimal ARDM prediction equals the sum of the reference model's prediction and the gradient of the value function. To train the model, VGPO samples trajectories online and regresses the model's intermediate predictions toward this theoretical optimum.

The contributions of this paper are as follows:

- We propose VGPO, a value-based reinforcement fine-tuning algorithm specifically designed for ARDMs. VGPO trains a value model to predict the total reward from partial trajectories and updates the diffusion score using the value model gradient.
- We apply VGPO to fine-tune DiTAR (Jia et al., 2025), a recently proposed state-of-the-art zero-shot TTS model based on ARDMs. We evaluate VGPO on two benchmark tasks: Task A, which aims to enhance the expressiveness of generated speech by optimizing the F0 variance (Quatieri, 2002). And task B, which focuses on improving TTS robustness when handling challenging long texts that are difficult for autoregressive TTS models.
- We propose to regularize VGPO through adversarial distribution matching (ADM). Unlike the common KL regularization (Fan et al., 2023), which struggles to correct errors accumulated during reinforcement fine-tuning, ADM effectively mitigates this issue.

The audio samples are available at https://vgpo-web.github.io/.

## 2 PRELIMINARIES

In this section, we describe the related background formulations necessary for describing our algorithm, including formulations of diffusion models (DM) and ARDMs.

### 2.1 NOTATIONS

Without loss of generality, we omit input conditions $\mathbf{c}$, such as text or prompt speech, and we assume that the generated token sequence has a fixed length $N$. Each sample trajectory $\mathbf{x} \in \mathbb{R}^{N\times d}$

consists of $N$ tokens $(x_1, \ldots, x_N)$, where each token resides in $\mathbb{R}^d$. We define $x_{\leq n}$ as the sequence $(x_1, \ldots, x_n)$, with similar definitions for $x_{<n}$, $x_{>n}$, and $x_{\geq n}$.

To prevent ambiguity between the diffusion time index, $t$, and the sequence index, $n$, we use a superscript on variables to denote the diffusion time when both $t$ and $n$ appear simultaneously (e.g., $x_n^t$). For conciseness, all $t$-conditioned models are written without the explicit time argument: $g(x_n^t) := g(x_n^t, t)$. This convention applies to all score models and value models.

## 2.2 AUTOREGRESSIVE DIFFUSION MODELS

**Diffusion Models.** One view that unifies many diffusion model formulations (Song et al., 2021b; Liu et al., 2023) is to view DMs as multiscale score estimators. For each diffusion time $t$, define a Gaussian transition distribution $q(x_t|x) := \mathcal{N}(x_t; \alpha_t x, \sigma_t^2 I_d)$, where $\alpha_t, \sigma_t > 0$. Suppose $p(x)$ is the clean token distribution. Then the marginal distribution of noisy tokens at time $t$ is:

$$p(x_t) := \int p(x) q(x_t|x) \mathrm{d}x. \tag{2}$$

A diffusion model trained with denoising score matching can be interpreted as a score estimator $s_\theta(x_t)$ that approximates the true score $\nabla \log p(x_t)$. Given the true score of all diffusion time $t$, one can draw samples from $p(x)$ with various diffusion model samplers such as DDPM (Ho et al., 2020), DDIM (Song et al., 2021a), SDE and ODE solvers (Song et al., 2021b).

**Autoregressive Models.** Let $p(\mathbf{x})$ denote the data distribution. By the chain rule of probability,

$$p(\mathbf{x}) = \prod_{n=1}^{N} p(x_n|x_{<n}), \tag{3}$$

An autoregressive (AR) model parameterizes these conditionals and approximates $p(x_n \mid x_{<n})$. Sampling proceeds ancestrally: first draw $x_1 \sim p(x_1)$, then $x_2 \sim p(x_2 \mid x_1)$, and, in general, $x_n \sim p(x_n \mid x_{<n})$ until all $N$ tokens are generated.

**Autoregressive Diffusion Models.** ARDMs sample from each conditional $p(x_n|x_{<n})$ with a diffusion model. Let $q(x_n^t|x_n) := \mathcal{N}(x_t; \alpha_t x, \sigma_t^2 I_d)$. For each diffusion time $t$, define the conditional marginal distribution $p(x_n^t|x_{<n})$ as:

$$p(x_n^t|x_{<n}) := \int p(x_n|x_{<n}) q(x_n^t|x_n) \mathrm{d}x_n \tag{4}$$

An ARDM learns a conditional multiscale score estimator $s_\theta(x_n^t|x_{<n})$ that estimates the conditional score $\nabla_{x_n^t} \log p(x_n^t|x_{<n})$, given clean tokens generated previously $x_{<n}$, and the token currently being denoised $x_n^t$. Let $\pi_\theta(\mathbf{x})$ be the sample distribution of the ARDM with the score model $s_\theta$. Assuming $s_\theta(x_n^t|x_{<n}) = \nabla_{x_n^t} \log p(x_n^t|x_{<n})$, and that the diffusion sampler is an exact solver, we have $\pi_\theta(\mathbf{x}) = p(\mathbf{x})$.

## 2.3 KL REGULARIZED REWARD MAXIMIZATION

**Definition 1** (KL RFT). Suppose that the distribution of the reference model is $\mu(\mathbf{x})$ and the reward function is $r(\mathbf{x}) : \mathbb{R}^{N \times d} \to \mathbb{R}$. Suppose that we pick the Kullback–Leibler (KL) divergence as our divergence regularizer, the goal of RFT now becomes the following:

$$\max_\rho \mathbb{E}_{\mathbf{x} \sim \rho(\mathbf{x})} [r(\mathbf{x})] - D_{\mathrm{KL}} (\rho(\mathbf{x}) \| \mu(\mathbf{x})). \tag{5}$$

**Theorem 1** (Solution of KL RFT). The closed form solution $\pi$ for the KL RFT problem in Definition (1) is given by:

$$\pi(\mathbf{x}) = \frac{\mu(\mathbf{x}) \exp r(\mathbf{x})}{Z}, \quad Z = \int \mu(\mathbf{x}) \exp r(\mathbf{x}) \mathrm{d}\mathbf{x}. \tag{6}$$

*Proof.* The proof can be found in previous works (Rafailov et al., 2023; Peng et al., 2019). A short proof is provided in Appendix A.1. □

## 3 METHODS

### 3.1 VALUE GUIDANCE FOR ARDMS

**Definition 2** (Soft Value Functions of ARDMs). Given a reward function $r(\mathbf{x}) : \mathbb{R}^{N \times d} \to \mathbb{R}$ and an ARDM with distribution $\mu(\mathbf{x})$. Soft value function $V(\cdot)$ estimates the future reward given partial information of the complete sample trajectory.

$$V(x_{\leq n}) := \log \left( \mathbb{E}_{\mu(x_{>n}|x_{\leq n})} \left[ \exp r(\mathbf{x}) \right] \right); \tag{7}$$

$$V(x_{<n}, x_n^t) := \log \left( \mathbb{E}_{\mu(x_n|x_n^t, x_{<n})} \left[ \exp V(x_{\leq n}) \right] \right) = \log \left( \mathbb{E}_{\mu(x_{\geq n}|x_n^t, x_{<n})} \left[ \exp r(\mathbf{x}) \right] \right). \tag{8}$$

**Theorem 2** (Solution of KL RFT for ARDMs). Given a reward function $r(\mathbf{x}) : \mathbb{R}^{N \times d} \to \mathbb{R}$ and a reference ARDM with distribution $\mu(\mathbf{x})$. Suppose $\pi(\mathbf{x}) \propto \mu(\mathbf{x}) \exp r(\mathbf{x})$ is the optimal solution of the KL RFT problem in Eq. (5). Then the conditional distribution $\pi(x_n^t|x_{<n})$ is given by:

$$\pi(x_n^t|x_{<n}) = \mu(x_n^t|x_{<n}) \exp \left( V(x_{<n}, x_n^t) - V(x_{<n}) \right). \tag{9}$$

where the soft value function $V(\cdot)$ is defined in Definition 2.

*Proof.* We provide two proofs for this result in A.2 and A.3. Proof in A.2 is based on directly solving for $\pi(x_n^t|x_{<n})$. We can view the ARDM sampling process as a Markov decision process (MDP), and apply MaxEnt RL (Levine, 2018) to the MDP. This leads to the proof in A.3. We show that $V(\cdot)$ is indeed the soft value function of the optimal policy. □

**Corollary 1** (Value Guidance). By applying $\nabla_{x_n^t} \log$ on both sides of Eq. (9), we can show that the conditional score of the optimal policy $\pi(x_n^t|x_{<n})$ is the sum of the reference score and value gradient:

$$\nabla_{x_n^t} \log \pi(x_n^t|x_{<n}) = \nabla_{x_n^t} \log \mu(x_n^t|x_{<n}) + \nabla_{x_n^t} V(x_{<n}, x_n^t). \tag{10}$$

### 3.2 ESTIMATING THE VALUE FUNCTION

Apply $\exp$ on both sides of Equation (8) gives:

$$\exp V(x_{<n}, x_n^t) = \mathbb{E}_{\mu(x_{\geq n}|x_{<n}, x_n^t)} \left[ \exp r(\mathbf{x}) \right]. \tag{11}$$

Therefore, given a parameterized soft value model $V_\phi(x_{<n}, x_n^t)$, we can approximate $V(x_{<n}, x_n^t)$ by minimizing the following Exp-MSE loss(Lu et al., 2023; Uehara et al., 2024):

$$\mathcal{L}_V^{n,t}(\phi) := \mathbb{E}_{\mu(\mathbf{x}), q(x_n^t|x_n)} \left[ \exp V_\phi(x_{<n}, x_n^t) - \exp r(\mathbf{x}) \right]^2. \tag{12}$$

The Exp-MSE loss in Eq. (12) is numerically unstable (Lu et al., 2023) due to the $\exp(\cdot)$ functions. We propose an alternative loss $\widehat{\mathcal{L}}_V^{n,t}(\phi)$ that shares the same global minimum, while providing better numerical stability. We leave the analysis in A.4. Previous works (Li et al., 2024b; Lu et al., 2023) observed that replacing the ***soft*** value function in Eq. (10) with the value function also provides good results, at the cost of losing the theoretical distribution guarantee provided by Theorem 1. We can minimize the following MSE loss to learn a value model $V_\phi(\cdot)$ approximating the value function.

$$\widetilde{\mathcal{L}}_V^{n,t}(\phi) := \mathbb{E}_{\mu(\mathbf{x}), q(x_n^t|x_n)} \left[ V_\phi(x_{<n}, x_n^t) - r(\mathbf{x}) \right]^2. \tag{13}$$

Algorithm 1 learns a (soft) value model given a reference ARDM model. As a result of Corollary 1, the value model can be used to guide ARDM sampling via Algorithm 2. Note that Algorithm 2 scales the value gradient by the parameter $\lambda$. Similar to classifier guidance (Ho & Salimans, 2022), we observe that setting $\lambda > 1$ can further improve the rewards of the samples.

---

**Algorithm 1** Value Training

---

**Require**: Reference ARDM model with distribution $\mu(\mathbf{x})$; initialized value model $V_\phi(\cdot)$

1: **while** $V_\phi$ has not converged **do**
2:      Sample a trajectory $\mathbf{x} \sim \mu(\mathbf{x})$
3:      Compute reward $r(\mathbf{x})$
4:      Update $\phi$ by minimizing the loss $\mathbb{E}_{n,t} \left[ \widehat{\mathcal{L}}_V^{n,t}(\phi) \right]$ or

       $\mathbb{E}_{n,t} \left[ \widetilde{\mathcal{L}}_V^{n,t}(\phi) \right]$
5: **end while**

---

---

**Algorithm 2** Value Guided Sampling

---

**Require**: Reference ARDM model with distribution $\mu(\mathbf{x})$ and score estimator $s_\mu(x_n^t|x_{<n})$; pre-trained value model $V_\phi(\cdot)$; guidance scale $\lambda \in \mathbb{R}$

1: **for** $n = 1$ to $N$ **do**
2:    Sample initial noise from $\mathcal{N}(0, I_d)$
3:    Sample $x_n$ by running diffusion sampler with score $s_\mu(x_n^t|x_{<n}) + \lambda \cdot \nabla_{x_n^t} V_\phi(x_{<n}, x_n^t)$ at $t$
4: **end for**

---

### 3.3 VALUE GUIDED POLICY OPTIMIZATION

Given an ARDM model with distribution $\pi_\theta(\mathbf{x})$ and score estimator $s_\theta(x_n^t|x_{<n})$, VGPO fine-tunes its score prediction to match the optimal solution in Eq. (10). Suppose the reference model $\mu(\mathbf{x})$ has score estimator $s_\mu(x_n^t|x_{<n}) = \nabla_{x_n^t} \log \mu(x_n^t|x_{<n})$, and let $V_\phi(x_{<n}, x_n^t)$ be a trained value model. The MSE loss $\mathbb{E}_{n,t}\left[\mathcal{L}_{\text{VD}}^{n,t}(\theta)\right]$ updates the prediction of $s_\theta$ to match the optimal solution $\nabla_{x_n^t} \log \mu(x_n^t|x_{<n}) + \nabla_{x_n^t} V_\phi(x_{<n}, x_n^t)$.

$$\mathcal{L}_{\text{VD}}^{n,t}(\theta) := \mathbb{E}_{\pi_{\text{sg}[\theta]}(\mathbf{x}),\, q(x_n^t|x_n)} \left\| s_\theta(x_n^t|x_{<n}) - \left(s_\mu(x_n^t|x_{<n}) + \nabla_{x_n^t} V_\phi(x_{<n}, x_n^t)\right) \right\|_2^2. \quad (14)$$

Note that the loss $\mathcal{L}_{\text{VD}}^{n,t}$ can be decomposed into two components that pull $s_\theta(x_n^t|x_{<n})$ in different directions. The first component is a KL regularization term (Liu et al., 2025a):

$$\mathcal{L}_{\text{KL}}^{n,t}(\theta) := \mathbb{E}_{\pi_{\text{sg}[\theta]}(\mathbf{x}),\, q(x_n^t|x_n)} \left\| s_\theta(x_n^t|x_{<n}) - s_\mu(x_n^t|x_{<n}) \right\|_2^2, \quad (15)$$

and the second is a value-guidance term:

$$\mathcal{L}_{\text{VG}}^{n,t}(\theta) := \mathbb{E}_{\pi_{\text{sg}[\theta]}(\mathbf{x}),\, q(x_n^t|x_n)} \left\| s_\theta(x_n^t|x_{<n}) - \text{sg}\left[s_\theta(x_n^t|x_{<n})\right] - \nabla_{x_n^t} V_\phi(x_{<n}, x_n^t) \right\|_2^2. \quad (16)$$

VGPO is described in Algorithm 3. In Algorithm 3 we multiply the term $\mathcal{L}_{\text{KL}}^{n,t}$ by $w_{\text{KL}}$ to control the KL regularization strength. Additionally, we can choose to update the value model online with Algorithm 1, turning the algorithm into a variant of online policy mirror decent (Tomar et al., 2020; Kimi Team et al., 2025; Ma et al., 2025).

---

**Algorithm 3** Value Guided Policy Optimization (VGPO)

---

**Require**: Reference ARDM model with distribution $\mu(\mathbf{x})$ and score estimator $s_\mu(x_n^t|x_{<n})$; pre-trained value model $V_\phi(\cdot)$; KL loss weight $w_{\text{KL}}$; target ARDM model $\pi_\theta(\mathbf{x})$ initialized from $\mu$ with score estimator $s_\theta(x_n^t|x_{<n})$
**Optionally Require**: Discriminator $D_\psi(\cdot)$; weight of adversarial gradient $w_{\text{A}}$

1: **while** $\pi_\theta$ has not converged **do**
2:    Sample a trajectory $\mathbf{x} \sim \pi_\theta(\mathbf{x})$
3:    Update $\theta$ by minimizing $w_{\text{KL}}\mathcal{L}_{\text{KL}}^{n,t}(\theta) + \mathcal{L}_{\text{VG}}^{n,t}(\theta)$ on randomly sampled $n, t$ pairs
4:    **if** enabled online value update **then**
5:       Invoke Algorithm 1 to sample from $\pi_\theta$ and update $V_\phi$
6:    **end if**
7:    **if** enabled adversarial distribution matching (Section 3.4) **then**
8:       Update $\theta$ by minimizing $w_{\text{A}}\mathcal{L}_{\text{G}}^{n,t}(\theta)$ on randomly sampled $n, t$ pairs
9:       Sample a trajectory $\mathbf{x}' \sim \mu(\mathbf{x})$
10:      Update $\psi$ by minimizing $\mathcal{L}_{\text{D}}^{n,t}(\psi)$ with $\mathbf{x}$ and $\mathbf{x}'$.
11:   **end if**
12: **end while**

---

### 3.4 REGULARIZING VGPO WITH ADVERSARIAL DISTRIBUTION MATCHING

There are several sources of gradient noise in Algorithm 3, including errors in value model prediction and noise from Monte Carlo loss estimation. We observed error accumulation running Algorithm 3, and find that the KL loss in Eq. (15) alone cannot fully rectify these errors. Tuning the KL weight $w_{\text{KL}}$ does not fully resolve this issue. When the KL weight is high, the original suboptimal behaviors

of the reference policy tend to be preserved in the target policy, leading to slow optimization. And when the KL weight is low, it fails to correct the error accumulation.

We propose an alternative regularization method based on token-level adversarial distribution matching (Goodfellow et al., 2014; Ho & Ermon, 2016; Huang et al., 2025). A discriminator network $D_\psi(x_n^t)$ is trained to distinguish between true noisy tokens and fake noisy tokens by minimizing $\mathbb{E}_{n,t}[\mathcal{L}_D^{n,t}(\psi)]$, where

$$\mathcal{L}_D^{n,t}(\psi) := \mathbb{E}_{\pi_{sg[\theta]}(\mathbf{x}), q(x_n^t|x_n)} \left[ D_\psi(x_n^t) + 1 \right]^2 + \mathbb{E}_{\mu(\mathbf{x}), q(x_n^t|x_n)} \left[ D_\psi(x_n^t) - 1 \right]^2. \quad (17)$$

And the generator is updated by minimizing $\mathbb{E}_{n,t}[\mathcal{L}_G^{n,t}(\psi)]$, where

$$\mathcal{L}_G^{n,t}(\theta) := \mathbb{E}_{\pi_{sg[\theta]}(\mathbf{x}), q(x_n^t|x_n)} \left\| s_\theta(x_n^t|x_{<n}) - sg\left[ s_\theta(x_n^t|x_{<n}) \right] - \nabla_{x_n^t} D_\psi(x_n^t) \right\|_2^2. \quad (18)$$

$\mathcal{L}_G$ can be obtained from replacing $V_\phi$ with $D_\psi$ in Eq. (16).

# 4 EXPERIMENTS

## 4.1 COMMON SETUP

**Base Model.** We fine-tuned a DiTAR model comprising approximately 0.4 billion parameters, trained on an internal dataset containing about 280k hours of Chinese and English speech. The language model (LM) within this architecture consists of 24 layers, while the diffusion head (DiT) includes 4 layers. All Transformer layers in the model have a hidden dimension of 1024, 16 attention heads, and a feed-forward network (FFN) dimension of 4096.

**Diffusion Sampler and CFG.** For all online and offline sampling in the experiments, we used the DDPM sampler with 16 sampling steps, with LM Guidance (Jia et al., 2025) weight $w = 2$. We always enable LM Guidance with $w = 2$ during training. See Appendix B.2 for more discussion about classifier-free guidance (CFG) in VGPO.

**VGPO Training.** All experiments were conducted on 32 A100 GPUs. We used the AdamW optimizer, with learning rate fixed to $1 \times 10^{-6}$, $\beta_1 = 0.9$, $\beta_2 = 0.95$, weight decay 0.01.

**Value Models.** All value models used in our experiments are initialized from the base DiTAR model, with the last linear layer of the diffusion head replaced by a zero initialized linear layer that outputs a scalar. We refer to the modified diffusion head as the value head. We choose to initialize the value head with diffusion head parameters, which brings better performance than random initialization in our preliminary experiments. The inputs to the value models are the same as in the DiTAR TTS model, including prompt speech, prompt text, and target text.

**Objective Evaluations.** We report the word error rate (**WER**) for speech intelligibility using Whisper-large-v3. For speaker similarity (**SIM**), we report the cosine similarity of speaker embeddings between prompts and generated audios, using a WavLM-TDCNN model. We computed WER and SIM using the same models and evaluation code[1] as in Seed-TTS (Anastassiou et al., 2024). For all evaluations, we run tests eight times and report the average. We also report the average KL loss (**KL**) on the test sets to report the divergence between the fine-tuned model and the base model. For evaluation, we used the KL loss definition in Eq. (17) in Liu et al. (2025b).

**Subjective Evaluations.** We evaluate the subjective quality of fine-tuned models using comparative mean opinion score (CMOS) for speech naturalness (N-CMOS), speaker similarity (S-CMOS), and speech expressiveness (E-CMOS). Human listeners compare the generated audio against base model response, and assign a score from $-2$ to $2$. See Appendix C for more details.

**Baselines.** We compare VGPO against the following baselines: (1) Guided sampling results using Algorithm 2. (2) ARDM-DPO, we evaluated the Liu et al. (2025b) (3) Base model sampling and best-of-K (BoK) sampling.

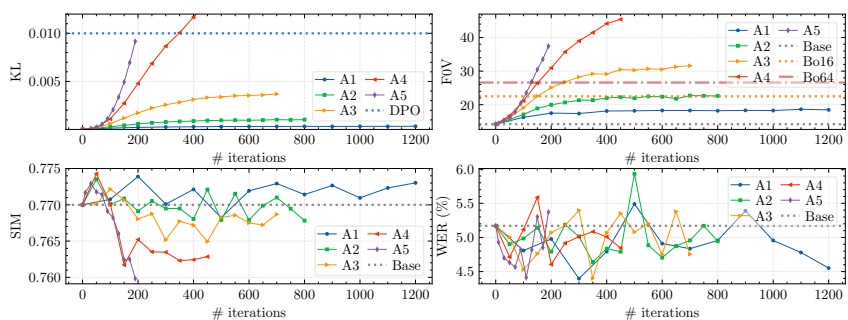

Figure 2: Evolution of KL, F0V, SIM, WER during VGPO training for task A.

## 4.2 TASK A: IMPROVING F0 VARIANCE

**Motivation.** We choose the fundamental frequency variance (F0V) as the reward for task A because it is a robust and label-free proxy of perceived expressiveness and directly counteracts monotony. F0V provides a simple and reproducible benchmark for testing the RFT algorithms of TTS models.

**Dataset.** We used the LibriTTS corpus (Zen et al., 2019) for both speech prompts and target texts, which includes 555 hours of recordings from 2,311 speakers. Evaluations were performed with 38 test cases, with prompts and target texts from 38 different speakers within the LibriTTS test-clean subset.

**Reward Function.** For a given utterance, we extract the F0 track of the voiced regions with an off-the-shelf F0 tracker. Then we apply a band-pass filter to the F0 track to focus on phrase-level variations. The reward is computed as the standard deviation of the filtered F0 track.

**Value Training.** We trained the value model directly with audios and transcripts from the LibriTTS corpus. The value model takes a transcript and an audio as input, and is trained to predict the F0V with the MSE loss. This is justified if we assume $\mu(\mathbf{x})$ is significantly similar to the dataset distribution. The value model is trained for 60k steps with a dynamic batch size of approximately 2 hours of speech per batch. The results of the value-guided sampling with this value model can be found in Table 1. The WER and SIM of the guided models are close to the base model, indicating good prior distribution preservation. We also report Best-of-N (BoN) results with $N \in \{16, 64\}$ in Table 1. We see that the value guidance can match the performance of Bo64 without reducing SIM.

Table 1: Selected objective evaluation results on task A.

| Method | F0V ↑ | WER ↓ | SIM ↑ | KL ↓ |
|---|---|---|---|---|
| Base | 14.2 | 5.17 | 0.770 | — |
| Best-of-16 | 22.5 | 4.74 | 0.770 | — |
| Best-of-64 | 26.6 | 4.93 | 0.770 | — |
| Guided $s = 2^2$ | 15.6 | 5.41 | 0.769 | — |
| Guided $s = 2^4$ | 19.3 | 5.24 | 0.771 | — |
| Guided $s = 2^5$ | 27.3 | 4.93 | 0.771 | — |
| Guided $s = 2^6$ | 52.6 | 5.76 | 0.760 | — |
| DPO$_{\beta = 200}^{200\,\text{steps}}$ | 29.2 | 3.73 | 0.765 | 0.010 |
| VGPO$_{A3}^{600\,\text{steps}}$ | 30.5 | 4.75 | 0.767 | 0.003 |

**VGPO Training.** During VGPO training, we sampled 32 pairs of prompts and target texts. And for each pair, we generate about 2 minutes synthesized speech with a dynamic number of rollouts per iteration.

Table 2: Subjective evaluation results for task A.

| Method | E-CMOS ↑ | N-CMOS ↑ | S-CMOS ↑ |
|---|---|---|---|
| DPO$_{\beta = 200}^{200\,\text{steps}}$ | $1.70 \pm 0.36$ | $-0.14 \pm 0.12$ | $-0.05 \pm 0.16$ |
| VGPO$_{A3}^{600\,\text{steps}}$ | $1.65 \pm 0.34$ | $0.05 \pm 0.13$ | $-0.03 \pm 0.20$ |

The evolution of KL, F0, and SIM during training can be found in Figure 2. Experiments A1, A2, A3, A4, A5 are trained with KL weight $2^{-4}, 2^{-5}, 2^{-6}, 2^{-7}, 2^{-8}$, respectively. When achieving similar F0V as the ARDM-DPO baseline, VGPO (600 steps, A3) obtains lower KL divergence, and higher SIM compared to DPO (200 steps, $\beta = 200$).

---

[1]https://github.com/BytedanceSpeech/seed-tts-eval

**Subjective Evaluations.** For each prompt in the test set, we generated three samples from the base model, DPO model, and VGPO model. We conducted N-CMOS, E-CMOS, and S-CMOS tests, for each we collected 114 scores. Evaluation results can be found in Table 2.

## 4.3 TASK B: ENHANCING ROBUSTNESS TO DIFFICULT TEXTS

Table 3: Selected objective evaluation results on task B. Table contains results of VGPO with different hyperparameters. O.V. is abbreviation for online value update. $w_{\mathrm{KL}}$ is the weight of the KL loss. $w_{\mathrm{A}}$ is the weight of ADM loss.

| ID (steps) | O.V. | $w_{\mathrm{KL}}$ | $w_{\mathrm{A}}$ | NLL $\downarrow$ | CER $\downarrow$ | SIM $\uparrow$ | KL $\downarrow$ |
|---|---|---|---|---|---|---|---|
| Base Model | — | — | — | 0.55 | 8.37 | 0.711 | 0 |
| Best-of-8 (WER) | — | — | — | 0.39 | 4.99 | 0.713 | — |
| Best-of-8 (NLL) | — | — | — | 0.27 | 6.79 | 0.712 | — |
| DPO$^{9000 \text{ steps}}_{\beta = 1600}$ | — | — | — | 0.32 | 6.32 | 0.712 | 0.009 |
| B1 | No | $2^{-11}$ | 0 | — diverged — | | | |
| B2 (6k) | No | $2^{-10}$ | 0 | 0.34 | 7.56 | 0.698 | 0.143 |
| B3 (6k) | No | $2^{-9}$ | 0 | 0.33 | 7.07 | 0.705 | 0.008 |
| B4 (6k) | Yes | $2^{-12}$ | 0 | 0.29 | 6.52 | 0.696 | 0.037 |
| B5 (6k) | Yes | $2^{-11}$ | 0 | 0.31 | 6.65 | 0.705 | 0.017 |
| B6 (6k) | Yes | $2^{-10}$ | 0 | 0.33 | 6.58 | 0.709 | 0.007 |
| B7 (6k) | Yes | $2^{-9}$ | 0 | 0.39 | 7.18 | 0.710 | 0.002 |
| B4 (15k) | Yes | $2^{-12}$ | 0 | 0.26 | 6.18 | 0.689 | 0.660 |
| B5 (15k) | Yes | $2^{-11}$ | 0 | 0.27 | 6.20 | 0.700 | 0.299 |
| B6 (15k) | Yes | $2^{-10}$ | 0 | 0.28 | 6.48 | 0.707 | 0.099 |
| B7 (15k) | Yes | $2^{-9}$ | 0 | 0.36 | 7.06 | 0.708 | 0.040 |
| B8 (15k) | Yes | 0 | $2^{-6}$ | 0.29 | 6.27 | 0.725 | 0.027 |
| B9 (15k) | Yes | 0 | $2^{-3}$ | 0.32 | 6.36 | 0.732 | 0.016 |

**Motivation.** Autoregressive TTS models often struggle to accurately read complex texts containing repetitive words or phrases. When evaluating our base model on such sentences, it frequently fails to correctly handle repetitions, either by omitting some repetitions, adding additional ones, or becoming trapped in a loop of repetitive generation, unable to terminate. In task B, our objective is to enhance the robustness of the DiTAR model when processing these challenging texts.

**Dataset.** We used the DiDiSpeech-2 (Guo et al., 2021) dataset as the source of speech prompts, which comprises 227 hours of recordings from 1,500 speakers. We excluded all speakers included in the hard subset of the SEED-TTS-Eval test set. For the training text set, we used a corpus of 100,000 long Chinese sentences. These sentences contain randomly repeated words, phrases, and clauses, making them difficult to synthesis correctly for autoregressive TTS models. All evaluations were performed on the test-hard subset of SEED-TTS-Eval. We excluded 2 test cases with the longest target texts, as they significantly exceed the context-length limit of our base model.

**Reward Modeling.** Following previous work (Du et al., 2025; Li et al., 2025), we utilize the likelihood of speech in automatic speech recognition (ASR) models as a proxy reward function. An alternative reward choice is CER, but computing CER is significantly slower than evaluating likelihood. For all experiments on task B, we employ a phoneme-based CTC model trained on DidiSpeech-2 as the reward model.

**Group Reward Normalization.** We find it necessary to normalize the reward for each pair of prompts and target texts to train the value model. Otherwise, the value model failed to capture the relative differences between trajectories with the same prompt and text. We perform a reward normalization similar to GRPO (Shao et al., 2024). Suppose that for each pair of prompt and target text, the CTC likelihoods are $\mathbf{r} = (r_1, \ldots, r_G) \in (0, 1)^G$ for $G$ samples. We normalize the reward to $\tilde{\mathbf{r}} \in \mathbb{R}^G$ as $\tilde{\mathbf{r}} := \frac{\mathbf{r} - \mathrm{mean}(\mathbf{r})}{\mathrm{std}(\mathbf{r})}$. This normalization is applied in all experiments of task B.

**Value Training.** To train the value model, we randomly generated approximately 430k pairs of prompts and target texts. For each pair, we produced 16 samples using the base DiTAR model, resulting in an offline corpus containing 27k hours of speech. The value model was trained on this dataset for 150k steps with MSE loss in Eq. (13), with a dynamic batch size of approximately 3 hours of generated speech per batch.

Table 4: Results of value guided sampling with different guidance scale $s$ on task B.

| $s$ | NLL $\downarrow$ | CER $\downarrow$ | SIM $\uparrow$ |
|---|---|---|---|
| 0 | 0.55 | 8.37 | 0.711 |
| 32 | 0.47 | 7.54 | 0.712 |
| 64 | 0.40 | **7.18** | 0.712 |
| 128 | 0.38 | 7.40 | 0.711 |
| 192 | 0.36 | 8.23 | 0.712 |

**VGPO Training.** During VGPO training, each iteration involved sampling 32 pairs of prompts and target texts, followed by generating approximately 8 rollouts for each pair. This process yielded approximately 2 hours of synthesized speech per training iteration. The value model (when updated online) and the discriminator (when ADM enabled) is trained 10 steps per iteration. The ARDM $\pi_\theta$ is always updated once per iteration.

**Results.** From the results in Table 4 and Tabel 3, we see that VGPO leads to better performance than value guided sampling in task B. We observe that enabling online value update leads to more stable training and better performance. VGPO with more iterations not only leads to a lower NLL and CER but also causes an increase in the accumulation of errors, as reflected in the decrease of SIM. For example, model B4 (15k) sometimes generates audio with audible artifacts such as sudden change of volume. We find that regularization with ADM can significantly mitigate this issue. Experiments B8 and B9 in Table 3 are initialized from B6 (10k), and further trained for 5k iterations with ADM enabled. For discriminator training, we used only the first 7 seconds of speech generated from the base model, as they include less distribution drift caused by exposure bias. As a result, B8 and B9 can beat the base model in SIM.

**Subjective Evaluations.** We randomly sampled 40 test cases from the test set and generated 3 random outputs each from the base, DPO, and VGPO models. For Task B, we conducted N-CMOS and S-CMOS evaluations, collecting 120 scores for each test. Evaluation results can be found in Table 5.

Table 5: Subjective evaluation results for task B.

| Method | N-CMOS $\uparrow$ | S-CMOS $\uparrow$ |
|---|---|---|
| DPO$^{9000 \text{ steps}}_{\beta = 1600}$ | $-0.03 \pm 0.11$ | $0.05 \pm 0.12$ |
| VGPO$^{15\text{k steps}}_{\text{B8}}$ | $-0.05 \pm 0.15$ | $0.19 \pm 0.23$ |

## 5 RELATED WORK

RFT algorithms for non-autoregressive (NAR) diffusion models (Uehara et al., 2024; 2025) and reinforcement learning methods with diffusion policies (Zhu et al., 2023) are closely related to our work. However, it is unclear whether these approaches can be effectively applied to fine-tune ARDMs. Existing research in both directions shares several core ideas: (1) Policy gradient (PG) methods (Black et al., 2024; Fan et al., 2023; Liu et al., 2025a; Xue et al., 2025) cast the diffusion sampling process as a Gaussian MDP. (2) Reward-weighted regression (RWR) methods (Lee et al., 2023; Zhang et al., 2024; Dong et al., 2023) iteratively maximize the likelihood of high-reward samples. (3) Direct gradient (DG) methods (Xu et al., 2023; Clark et al., 2024; Li et al., 2025) differentiate through the diffusion sampling process to optimize the reward. (4) Multiple works adapt DPO to diffusion models (Wallace et al., 2024; Yang et al., 2024; Liu et al., 2025b).

## 6 CONCLUSION

We introduce Value-Guided Policy Optimization (VGPO), an online actor–critic reinforcement fine-tuning algorithm tailored for autoregressive diffusion models. We derive VGPO from the exact solution to KL-regularized policy optimization, thereby providing strong theoretical guarantees. We apply VGPO to fine-tune the DiTAR TTS model and evaluate it on two benchmarks. Empirically, VGPO exhibits stable training and achieves better results than DPO. For future work, VGPO can be applied to fine-tuning image and video ARDMs. In this work, we focus on an ARDM architecture with a causal transformer and a diffusion head. It would be interesting to investigate whether VGPO generalizes to other variants of autoregressive diffusion models, including masked autoregressive diffusion models (Li et al., 2024a) and diffusion-forcing models (Song et al., 2025).

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

## A   ADDITIONAL DERIVATIONS

### A.1   OPTIMAL SOLUTION FOR KL-CONSTRAINED RFT OBJECTIVE IN EQ. (5)

*Proof.* The objective in Eq. (5) can be written as

$$
\begin{aligned}
\mathcal{J} &= \mathbb{E}_{\mathbf{x} \sim \rho(\mathbf{x})} \left[ r(\mathbf{x}) - \log \frac{\rho(\mathbf{x})}{\mu(\mathbf{x})} \right] \\
&= \mathbb{E}_{\mathbf{x} \sim \rho(\mathbf{x})} \left[ -\log \frac{Z}{\exp r(\mathbf{x})} - \log \frac{\rho(\mathbf{x})}{\mu(\mathbf{x})} \right] + \text{const.} \\
&= \mathbb{E}_{\mathbf{x} \sim \rho(\mathbf{x})} \left[ -\log \frac{\rho(\mathbf{x})}{\pi(\mathbf{x})} \right] + \text{const.} = -D_{\text{KL}}(\rho(\mathbf{x}) \| \pi(\mathbf{x})) + \text{const.},
\end{aligned}
\tag{19}
$$

where $Z$ and the optimal policy $\pi(\mathbf{x})$ are defined in Eq. (6). $\qquad\square$

### A.2   PROOF OF THEOREM 2 BASED ON SOLVING FOR $\pi(x_n^t | x_{<n})$

*Proof.* From Theorem 1 we know that $\pi(\mathbf{x}) = \mu(\mathbf{x}) \exp r(\mathbf{x})/Z$. We can find the relation between $\pi(x_{\leq n})$ and $\mu(x_{\leq n})$ by marginalizing all $x_{>n}$.

$$
\begin{aligned}
\pi(x_{\leq n}) &= \int \mu(\mathbf{x}) \frac{\exp r(\mathbf{x})}{Z} \mathrm{d}x_{>n} = \int \mu(x_{>n}|x_{\leq n}) \mu(x_{\leq n}) \frac{\exp r(\mathbf{x})}{Z} \mathrm{d}x_{>n} \\
&= \mu(x_{\leq n}) \cdot \frac{1}{Z} \cdot \int \mu(x_{>n}|x_{\leq n}) \exp r(\mathbf{x}) \mathrm{d}x_{>n} = \mu(x_{\leq n}) \frac{\exp V(x_{\leq n})}{Z}.
\end{aligned}
\tag{20}
$$

Our goal is to obtain the relation between $\pi(x_n|x_{<n})$ and $\mu(x_n|x_{<n})$. Divide both sides of Eq. (20) with $\pi(x_{<n})$ gives:

$$
\frac{\pi(x_{\leq n})}{\pi(x_{<n})} = \pi(x_n|x_{<n}) = \frac{\mu(x_{\leq n}) \exp V(x_{\leq n})}{\mu(x_{<n}) \exp V(x_{<n})} = \mu(x_n|x_{<n}) \frac{\exp V(x_{\leq n})}{\exp V(x_{<n})}.
\tag{21}
$$

Following Lu et al. (2023), we multiply both sides of Eq. (21) with $q(x_n^t|x_n)$ and marginalize $x_n$, which gives:

$$
\begin{aligned}
\pi(x_n^t|x_{<n}) &= \int \pi(x_n|x_{<n}) q(x_n^t|x_n) \mathrm{d}x_n = \int q(x_n^t|x_n) \mu(x_n|x_{<n}) \frac{\exp V(x_{\leq n})}{\exp V(x_{<n})} \mathrm{d}x_n \\
&= \mu(x_n^t|x_{<n}) \cdot \frac{1}{\exp V(x_{<n})} \cdot \int \mu(x_n|x_n^t, x_{<n}) \exp V(x_{\leq n}) \mathrm{d}x_n \\
&= \mu(x_n^t|x_{<n}) \frac{\exp V(x_{<n}, x_n^t)}{\exp V(x_{<n})}.
\end{aligned}
\tag{22}
$$

Now apply logarithm on both sides of Eq. (22) and take the gradient with respect to $x_n^t$ gives our result in Eq. (10). $\qquad\square$

### A.3   PROOF OF THEOREM 2 BASED ON MAXIMUM ENTROPY RL

In this section, we analyze diffusion models and ARDMs using the discrete-time sampler introduced in DDPM Ho et al. (2020). This choice does not limit our results to the DDPM sampler, as will become clear from the following derivation. The following derivation are based on the Markov chain view of ARDMs. See Liu et al. (2025b, Fig. 1).

### A.3.1 Background: Maximum Entropy RL

Consider a Markov Decision Process (MDP) in which all trajectories consist of $T$ steps. Each trajectory can be expressed as follows:

$$\tau := (s_0, a_0, s_1, a_1, \ldots, s_T), \tag{23}$$

where $s_T$ represents a terminal state. Given a reference policy $\mu(a|s)$ and a target policy $\pi(a|s)$, Maximum Entropy Reinforcement Learning (MaxEnt RL) (Levine, 2018) optimizes the following objective:

$$\pi^* := \arg\max_\pi \mathbb{E}_\pi \left[ r(s_T) - \sum_{t=0}^{T-1} \mathrm{D_{KL}}(\pi(\cdot|s_t)\|\mu(\cdot|s_t)) \right], \tag{24}$$

where $\mathrm{D_{KL}}$ denotes the Kullback-Leibler divergence.

The soft state value function, $V^\pi$, is defined as:

$$V^\pi(s_t) := \mathbb{E}_\pi \left[ r(s_T) - \sum_{u=t}^{T-1} \mathrm{D_{KL}}(\pi(\cdot|s_u)\|\mu(\cdot|s_u)) \mid s_t \right]. \tag{25}$$

Similarly, the soft Q function, $Q^\pi$, is defined as:

$$Q^\pi(s_t, a_t) := \mathbb{E}_{s_{t+1} \sim p(s_{t+1}|s_t, a_t)} \left[ V^\pi(s_{t+1}) \right], \quad t < T, \tag{26}$$

where $p(s_{t+1}|s_t, a_t)$ represents the transition dynamics of the environment.

For the optimal policy $\pi^*$, along with the corresponding optimal value functions $V^*$ and $Q^*$, the following expression holds (Levine, 2018):

$$\pi^*(a_t|s_t) := \mu(a_t|s_t) \cdot \frac{\exp(Q^*(s_t, a_t))}{\exp(V^*(s_t))}. \tag{27}$$

Additionally, the optimal soft value function $V^*$ satisfies the following:

$$V^*(s_t) = \log \int \mu(a_t|s_t) \cdot \exp(Q^*(s_t, a_t)) \, \mathrm{d}a_t. \tag{28}$$

With the above results from MaxEnt RL, we are ready to prove Theorem 2.

*Proof.* The Markov sampling chain of ARDMs can be embedded in an MDP. The state space and initial distribution of the MDP are the same as those of the Markov chain in Liu et al. (2025b, Fig. 1). In state $s_n^t = (x_{1:n}^0, x_n^t)$, where $t > 1$, the action is $x_n^{t-1}$, determined by the ARDM. Given action $x_n^{t-1}$ in state $s_n^t$, the MDP deterministically transitions to $(x_{1:n}^0, x_n^{t-1})$. In state $s_n^0$, where $n < N$, the action is a randomly sampled Gaussian noise $x_{n+1}^T$. Given action $x_{n+1}^T$ in state $s_n^0$, the MDP deterministically transitions to $(x_{1..n}^0, x_{n+1}^T)$. Intermediate states and actions in the MDP do not receive a reward; only the terminal state receives a reward $r(x_{1..N}^0)$.

Since we have mapped the ARDM sampling Markov chain as an MDP, we can analyze it using MaxEnt RL. For ease of description, we will first flatten the sampling Markov chain of ARDM by identifying state $s_{nT-t}$ with state $s_n^t$. The Markov chain of ARDM sampling can now be written as $s_0 \rightarrow s_1 \rightarrow \cdots \rightarrow s_{NT}$. First, since the MDP has deterministic transitions, we have

$$V^*(s_{u+1}) = Q^*(s_u, a_u) \tag{29}$$

for all possible pairs $(s_u, a_u, s_{u+1})$. Suppose that the reference policy is $\mu(s_{u+1}|s_u)$, the optimal policy $\pi^*(s_{u+1}|s_u)$ satisfies:

$$\pi^*(s_{u+k}|s_u) \cdot \exp(V^*(s_u)) = \mu(s_{u+k}|s_u) \cdot \exp(V^*(s_{u+k})), \tag{30}$$

Rewriting Equation 30 by expanding the states as tokens, we have the following.

$$\pi^*(x_n^t|x_{1:n}^0) = \mu(x_n^t|x_{1:n}^0) \cdot \frac{\exp(V^*(x_{1:n}^0, x_n^t))}{\exp(V^*(x_{1:n}^0))}. \tag{31}$$

$\square$

## A.4 Numerical Stability of Exp-MSE Loss

The Exp-MSE loss in Eq. (12) is numerically unstable during optimization. We propose an alternative loss that shares the same global minimum, while providing better numerical stability. Here, sg is the stop gradient operator.

$$\widehat{\mathcal{L}}_V^{n,t}(\phi) := \mathbb{E}_{\mu(\mathbf{x}),q(x_n^t|x_n)} \Big[ \exp\big(V_\phi(x_{<n}, x_n^t) - \mathrm{sg}\big[V_\phi(x_{<n}, x_n^t)\big]\big) - \\ \exp\big(r(\mathbf{x}) - \mathrm{sg}\big[V_\phi(x_{<n}, x_n^t)\big]\big) \Big]^2. \tag{32}$$

*Proof.* To see why is this the case, consider the following simplified version of the Exp-MSE loss. Here $f_\theta(x)$ is a neural network estimator, and $r(x, y)$ is an arbitrary bounded function depending on both $x, y$. The conditional distribution of $y$ given $x$ is $p(y|x)$.

$$\ell_\theta(x) := \int p(y|x)\big(\exp f_\theta(x) - \exp r(x, y)\big)^2 \, \mathrm{d}y. \tag{33}$$

The gradient has high variance since it contains multiple exponential functions:

$$\nabla_\theta \ell_\theta(x) = \int p(y|x)\Big[2\big(\exp f_\theta(x) - \exp r(x, y)\big)\exp f_\theta(x)\Big]\nabla_\theta f_\theta(x) \, \mathrm{d}y. \tag{34}$$

We propose the following surrogate loss:

$$\hat{\ell}_\theta(x) := \int p(y|x)\Big(\exp\big(f_\theta(x) - \mathrm{sg}[f_\theta(x)]\big) - \exp\big(r(x, y) - \mathrm{sg}[f_\theta(x)]\big)\Big)^2 \mathrm{d}y. \tag{35}$$

The gradient of this surrogate loss is

$$\nabla_\theta \hat{\ell}_\theta(x) = 2\left(1 - \int p(y|x)\exp\big(r(x, y) - f_\theta(x)\big) \, \mathrm{d}y\right)\nabla_\theta f_\theta(x). \tag{36}$$

This loss has the desired unique global minimum:

$$f_\theta^*(x) \;=\; \log \int p(y|x)\exp r(x, y)\mathrm{d}y. \tag{37}$$

$\square$

# B More Implementation Details

## B.1 Discriminator Architecture in Task B

Adversarial distribution matching is applied to Task B VGPO training. The discriminator in our experiments is initialized from the base DiTAR model. The LM in DiTAR is adapted as a prompt encoder, taking prompt speech as input and outputting a speaker embedding. The diffusion head is adapted into a discriminator network that takes a speaker embedding, a noisy token, and its diffusion time as input. Similar to the value model, the discriminator is first trained on offline sampled trajectories from the base model.

## B.2 Classifier-Free Guidance in VGPO

Many ARDMs rely on classifier-free guidance (CFG) to obtain high-quality samples (Song et al., 2025; Sun et al., 2024; Jia et al., 2025). However, CFG violates the assumption that $q(x_n^t|x_n)$ can approximate the true marginal distribution $p(x_n^t|x_n)$, creating a gap between theory and practice. In our implementation, we consistently enabled LM guidance with a fixed weight $w = 2$ in DiTAR, which we found to work well. Consequently, the diffusion head is evaluated twice in each inference and training step. Because the diffusion head in DiTAR is lightweight, the additional computational cost remains acceptable.

## C  SUBJECTIVE EVALUATIONS

We conducted comparative mean opinion score (CMOS) tests to evaluate the performance of fine-tuned models against the base model. The evaluation user interfaces for the different CMOS tests are shown in Figures 3, 4, and 5. In all tests, listeners are presented with a set of instructions, the target text, and audio generated by two models. For the speaker similarity test, the prompt audio is provided as the reference.

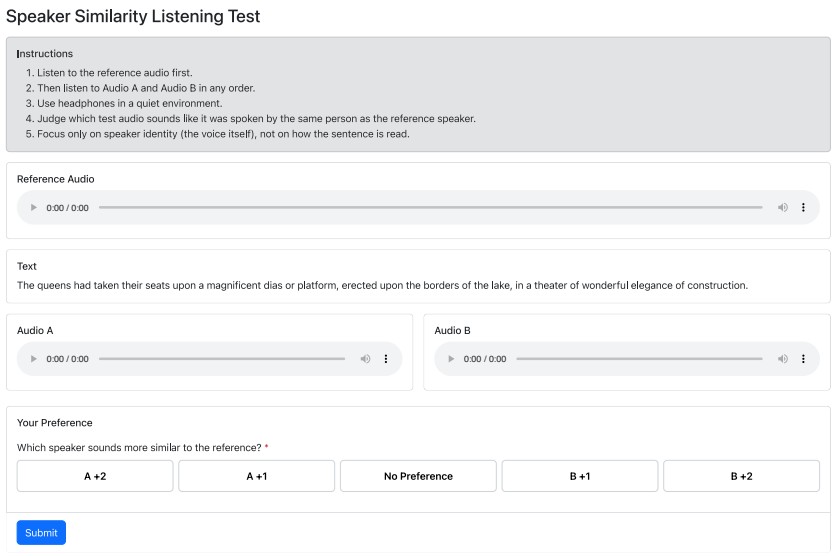

Figure 3: Screen shot of CMOS evaluation interface for speech naturalness (N-CMOS).

Figure 4: Screen shot of CMOS evaluation interface for speaker similarity (S-CMOS).

## D  LLM USAGE

LLMs including GPT-5 and GPT-4o are used for correcting grammar errors.

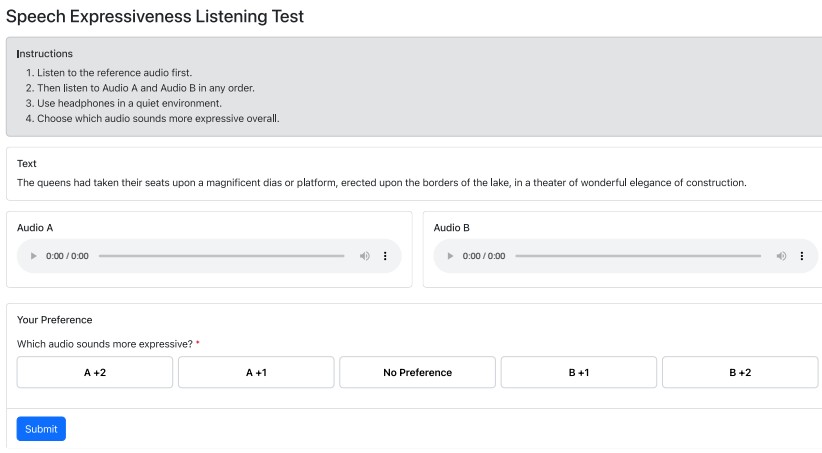

Figure 5: Screen shot of CMOS evaluation interface for speech expressiveness (E-CMOS).

