# OpenReview forum: "VGPO: Fine-Tuning Speech Autoregressive Diffusion Models with Value Guided Policy Optimization"
_ICLR.cc/2026/Conference — Submitted to ICLR 2026_

### Official Review · Reviewer_jkmb · 2025-10-28

**Soundness:** 3
**Presentation:** 2
**Contribution:** 3
**Rating:** 4
**Confidence:** 4

**Summary:**

This paper proposes Value-Guided Policy Optimization (VGPO), an online, actor-critic reinforcement learning algorithm designed to fine-tune speech autoregressive diffusion models (ARDMs). The authors provide a theoretical derivation, based on KL-regularized reward maximization, showing that the optimal policy's score is the sum of the reference model's score and the gradient of a "soft value function." Based on this, VGPO trains a value model (critic) to predict future rewards and updates the ARDM (actor) policy to match this theoretically optimal, value-guided score. The method is evaluated on two TTS tasks: improving F0 variance (expressiveness) and enhancing robustness to difficult, repetitive text. To combat error accumulation, the authors also introduce an Adversarial Distribution Matching (ADM) regularizer.

**Strengths:**

The paper tackles a novel and significant problem: how to apply reinforcement fine-tuning (RFT) to the SOTA class of autoregressive diffusion models for speech. This is a non-trivial challenge, and a robust solution would be a valuable contribution.

The derivation in Section 3.1, which connects the KL-RFT objective to a concrete update rule for the diffusion score (Corollary 1), is clean and provides a theoretical basis for the algorithm.

The method shows clear empirical gains on two distinct and relevant tasks. VGPO outperforms baselines like Best-of-K and an ARDM-DPO variant in both objective and subjective metrics. The identification of error accumulation as a key problem is also a good insight.

**Weaknesses:**

Significant Gap Between Theory and Practice: The entire theoretical derivation (Sec 3.1) is based on optimizing a soft value function (Def 2), which requires the unstable Exp-MSE loss (Eq 12). However, the paper states in Sections 4.2 and 4.3 that the experiments actually use the standard MSE loss (Eq 13), which estimates the "hard" value (expected reward). This is a major disconnect. The theoretical guarantees of Corollary 1, which are the paper's main theoretical anchor, do not apply to the algorithm that was actually implemented and tested. This gap is not adequately justified or analyzed.

Omission of Computational Cost: VGPO is an online, on-policy algorithm, which requires new rollouts from the policy at every training iteration. This is in sharp contrast to the offline DPO baseline. Online RFT is known to be extremely computationally expensive and sample-inefficient. The paper provides no discussion whatsoever on the computational cost (e.g., training time, GPU hours, sample-efficiency) of VGPO versus its baselines.

Insufficient Baseline Comparison: The comparison to ARDM-DPO (Liu et al., 2025b) is a key result, but the implementation details are too sparse. DPO requires preference pairs (chosen, rejected). The paper does not explain how these pairs were generated for the DPO baseline (e.g., were they sampled and ranked by the reward function?). Without this, the comparison is not reproducible and it's difficult to assess whether the DPO baseline was implemented in its strongest form.

**Questions:**

For both Task A (F0V) and Task B (NLL), how exactly were the preference pairs for the DPO baseline generated?
For example, did you sample two candidates from the base model and then use the reward function (F0V or NLL) to label them as 'chosen' and 'rejected'?
What β (beta) values were swept for the DPO baseline, and how was the final β (e.g., β=200 in Task A) selected? This is crucial for ensuring a fair comparison.

The Adversarial Distribution Matching (ADM) regularizer was shown to be critical in Task B for mitigating speaker similarity degradation (Table 3, models B8/B9). This is a very interesting and valuable finding. However, its application seems specific to Task B.
Was ADM also applied to Task A? If so, what was its effect?
If ADM was not used for Task A, why was it unnecessary? Did Task A not suffer from the same "error accumulation" and distribution drift?
Does this suggest that the standard KL regularization in VGPO (Eq. 15) is generally insufficient for complex tasks or long training runs? When should a practitioner using VGPO choose to enable ADM?

In Section 4.3, you state that "Group Reward Normalization" was "necessary to train the value model" for Task B. This seems like a critical, non-obvious implementation detail.
Could you elaborate on what failed when this normalization was not used? Did the value loss not converge, or did it converge to a useless "average" value?
Was this normalization also applied in Task A, or was it specific to the CTC likelihood reward in Task B?

---

> ### Author Response · Authors · 2025-12-02
>
> **Theoretical Gap (Value v.s. Soft Value)**: Training the value model with the surrogate loss provided in Appendix A.4 is numerically stable and more theoretically consistent. However, we observed little difference in results when training with MSE versus the proposed surrogate loss in preliminary experiments. Therefore, we recommend simply using the MSE loss for simplicity.
>
> **Details for the DPO Baseline**: All implementation details of the DPO baseline can be found in [1], including the specific methods for generating preference pairs. We conducted a grid search on $\beta$ and selected the best results for the baseline comparison.
>
> **Questions about ADM**: ADM was not applied to Task A, as VGPO training is relatively short for this task and we did not observe significant error accumulation. We found standard KL regularization insufficient for long training runs; therefore, we recommend using ADM specifically when error accumulation is observed and tuning KL fails to resolve the issue.
>
> **Questions about Group Reward Normalization**: In Task A, normalization is not necessary, as the value model can fit the F0V almost perfectly (MSE < 1.0). However, in Task B, we observed that NLL is much harder for the value model to predict. Without normalization, the model converged to predicting nearly identical NLL values for all rollouts using the same prompt.
>
> **References**
>
> [1] Liu, Z., et al. (2025). Direct Preference Optimization for Speech Autoregressive Diffusion Models. arXiv:2509.18928.

---

### Official Review · Reviewer_vauG · 2025-11-01

**Soundness:** 2
**Presentation:** 3
**Contribution:** 2
**Rating:** 4
**Confidence:** 4

**Summary:**

The paper introduces VGPO (Value-Guided Policy Optimization), an actor–critic reinforcement fine-tuning (RFT) algorithm tailored to Autoregressive Diffusion Models (ARDMs) for speech/TTS. A causal value model predicts expected future rewards given partial trajectories; the actor (ARDM) is updated using the value gradient, which the authors argue yields an optimal update that equals the reference model’s prediction plus the value gradient (under a MaxEnt-RL/energy-guidance view). The method is evaluated by fine-tuning DiTAR on two targets: (A) increasing F0 variance to enhance expressiveness; (B) improving text log-probability for robustness on long/hard text. An additional Adversarial Distribution Matching (ADM) regularizer replaces conventional KL to mitigate drift. Experiments report improved expressiveness/robustness while preserving naturalness and speaker similarity

**Strengths:**

1. The paper takes a relatively unexplored direction by applying reinforcement learning—specifically an actor–critic framework—to autoregressive diffusion models. This adaptation is nontrivial and provides a promising pathway toward controllable fine-tuning in continuous latent ar models.

2.  The paper is well-written and easy to follow. The theoretical formulation is consistent and mathematically coherent.

**Weaknesses:**

The current experiments and motivations are too narrowly scoped to the text-to-speech (TTS) setting, focusing mainly on F0 variance (expressiveness) and robustness to long text. While these are valid and practically meaningful objectives, they are very specific to TTS and do not convincingly demonstrate that the proposed VGPO framework offers a general contribution to the broader class of autoregressive diffusion models.

For an ICLR-level paper, one would expect more general insights or cross-domain validation. For instance, showing that the same value-guided optimization can improve the ar diffusion model on image generation [1]

At present, the work feels too task-specific, and its novelty risks being interpreted as a domain-engineering improvement more appropriate for a venue such as Interspeech, rather than ICLR. Expanding the experimental scope to include at least one non-speech autoregressive diffusion task would greatly improve the paper’s impact and positioning.

[1] Li T, Tian Y, Li H, et al. Autoregressive image generation without vector quantization[J]. Advances in Neural Information Processing Systems, 2024, 37: 56424-56445.

**Questions:**

It is well known that directly using GRPO to optimize a specific metric can improve that metric’s performance — but likewise, the generated quality in other dimensions may degrade. Such optimization may lead to reward hacking. Does VGPO fall into a similar risk?

---

> ### Author Response · Authors · 2025-12-02
>
> **Risk of Reward Hacking**: We confirm that, like other RL methods (e.g., GRPO), VGPO is susceptible to reward hacking. If the reward signal is maximized without constraint, the model may exploit the metric at the expense of other qualitative aspects. To counter this, we explicitly incorporate regularization terms—specifically the KL divergence (or the proposed Adversarial Distribution Matching, ADM).

---

### Official Review · Reviewer_BGr3 · 2025-11-02

**Soundness:** 3
**Presentation:** 3
**Contribution:** 3
**Rating:** 6
**Confidence:** 3

**Summary:**

Diffusion models use a continuous latent space, and using RL to directly optimize user-defined rewards remains an open challenge. Instead of saying PPO/DPO are unstable, the paper shows that KL alone cannot fully correct error accumulation during reinforcement fine-tuning.
This paper proposes VGPO (Value-Guided Policy Optimization), a reinforcement learning method designed to overcome these issues in diffusion-based TTS models. VGPO adds a value function inside the diffusion model and uses its value-gradient to guide each generation step. The paper also introduces Adversarial Distribution Matching (ADM) as a new regularization method, making the training process more stable than using only KL divergence.
Experimental results show that VGPO improves both expressiveness and robustness compared to DPO. Overall, the paper shows that VGPO helps reduce the instability of reinforcement learning in diffusion-based models.

**Strengths:**

This paper shows that reinforcement learning can be applied to diffusion models and presents a policy gradient method suitable for their continuous and non-differentiable nature. It demonstrates that gradients can be passed through the value function, making fine-grained optimization possible. The value function is also well formulated for continuous diffusion processes. They further show that the reward function using F0 variance and ASR likelihood converges stably during training.

**Weaknesses:**

[1] The paper does not describe key settings for the discriminator, such as learning rate, update frequency, or stabilization methods.
Because of this, the stability of ADM training and reproducibility remain uncertain.
[2] To answer the question “Does the improvement come only from reward tuning?”,
the paper should compare results without value guidance or without ADM.
This would show how much each component contributes to the overall performance.
[3] Even though the rewards and quantitative metrics improved, it remains unclear how the generated speech is perceptually better, since only a small-scale CMOS test was conducted and no qualitative analysis or listening examples were discussed.

**Questions:**

[1] Can you provide more details about the ADM discriminator training, such as learning rate, update frequency, or any stabilization methods used? This information would make the training process clearer and easier to reproduce.
[2] Did you run experiments without value guidance or without ADM? Showing these results would help understand how much each part contributes to the final performance.
[3] How is the generated speech perceptually better? Since only a small-scale CMOS test was shown, some listening examples or qualitative analysis would make the perceptual claim more convincing.

---

> ### Author Response · Authors · 2025-12-02
>
> **ADM Discriminator Training Details**: We appreciate the opportunity to clarify the training configuration. As noted in the manuscript (lines 303 and 444), the ADM discriminator shares the same optimizer settings (including learning rate and scheduler) as the value model and the ARDM policy. We did not find it necessary to apply specific regularization techniques or distinct update frequencies; the standard adversarial training setup proved stable throughout our experiments without signs of collapse or oscillation.
>
> **Component Contribution (Ablation Studies)**: To isolate the contributions of Value Guidance versus ADM, we provide the following breakdown: VGPO without ADM: Results for this configuration on Task B are already detailed in Table 3 of the manuscript. VGPO without Value Guidance (ADM Only): We performed an additional ablation where both Value Guidance and the KL loss were disabled, leaving only the ADM loss active for Task B. This setting achieved a Character Error Rate (CER) of 7.90% and Speaker Similarity (SIM) of 0.731. This is only a marginal improvement over the Base Model (CER 8.37%, SIM 0.711). Conclusion: Comparing the full VGPO method against the "ADM Only" result demonstrates that Value Guidance is the primary driver of the significant performance gains, while ADM serves as an effective regularizer to maintain distribution matching.
>
> **Perceptual Quality**: We invite the reviewer to examine the qualitative improvements via our anonymous online supplement: https://vgpo-web.github.io/.

---

### Official Review · Reviewer_1DeF · 2025-11-03

**Soundness:** 2
**Presentation:** 3
**Contribution:** 1
**Rating:** 2
**Confidence:** 3

**Summary:**

The paper proposes a value-guided policy optimization method for autoregressive diffusion models and evaluates it on two different objectives for a speech synthesis model, namely F0 variance for improved naturalness and text probability optimization for addressing issues regressive models have with long text. A soft value function, which predicts the future reward given partial trajectory information, is learned from a reference model and a target reward function (which is an approach related to classifier guidance, except that the reward here is a soft value instead of a hard classification value).

**Strengths:**

The paper proposes a novel method for RL training for autoregressive diffusion models, which allows one to incorporate specific reward functions to improve certain characteristics of the generated output. The proposed method is justified theoretically and supported by proofs (although I did not review the proofs myself).

**Weaknesses:**

The objective metrics measured in the experiments are unrelated to one of the experiments (F0 variability/naturalness), although that is partially covered by the subjective metrics. The authors present F0V as being "the higher the better", but that is not objectively true: speech with very high F0V is perceived as very unnatural. The subjective metrics do not seem to present a convincing case for VRPO vs. DPO, given that there is basically no significant change in all scores for both experiments.

I would have liked to see an ablation study showing the effect of each decision in designing the proposed method, instead of just baselines with DPO and best-of-K sampling.

**Questions:**

Why was the model only evaluated on TTS? It would have been interesting to see the effect of the proposed optimization strategy in other domains.

---

> ### Author Response · Authors · 2025-12-02
>
> **Potential Reward Hacking**: We acknowledge that F0 Variance (F0V) is not strictly "higher is better." Designing reward functions to optimize speech naturalness remains a challenging open problem. We utilized Task A (F0V) primarily as an accessible benchmark to evaluate the stability and efficiency of reinforcement fine-tuning algorithms rather than as a direct proxy for perceptual quality. We recognize that several dedicated reward models for text-to-speech have been proposed recently and plan to benchmark our algorithms using these models in future work.
>
> **VGPO vs. DPO**: Regarding Task A, VGPO demonstrates superior training stability across a wider range of KL weights compared to the baseline. In contrast, we observed that DPO [1] leads to a deterioration in speaker similarity (SIM) as training progresses, necessitating careful tuning of training steps and hyperparameters. VGPO mitigates this instability, offering a more robust alternative.
>
> **References**
>
> [1] Liu, Z., et al. (2025). Direct Preference Optimization for Speech Autoregressive Diffusion Models. arXiv:2509.18928.

---

### Meta-Review · Area_Chair_13at · 2026-01-07

**Summary:**

Several critical concerns led to a borderline/reject consensus.

Narrow Experimental Scope:  The evaluation is strictly limited to the TTS domain. The lack of cross-domain validation (e.g., image generation) makes it difficult to assess the framework’s generalizability to the broader class of AR diffusion models.

Gap Between Theory and Practice: A significant disconnect exists between the theoretical derivation (which relies on a "soft" value function and Exp-MSE loss) and the actual implementation (which uses standard MSE and "hard" value estimation). This undermines the strength of the paper's theoretical contributions.

Metric Validity and Reward Hacking: The use of F0V as a proxy for naturalness, arguing that higher F0V does not inherently mean better quality. The subjective improvements over the DPO baseline were also perceived as marginal.

Efficiency and Cost: As an online RL method, VGPO’s computational overhead compared to offline alternatives like DPO was not adequately discussed or quantified.

**Reviewer Concerns:**

Addressed:

Ablation Studies and Implementation Details: The authors provided missing details for the Adversarial Distribution Matching (ADM) discriminator and included an ablation study isolating the effects of Value Guidance vs. ADM. They also clarified the necessity of "Group Reward Normalization" for stable training.

DPO Baseline Clarity: The authors clarified how preference pairs were generated for the DPO comparison and the hyperparameter tuning involved.

Perceptual Evidence: An anonymous demo website was provided to showcase qualitative improvements, addressing the limitations of the small-scale CMOS test.

Outstanding

Generalization: The authors did not provide any non-speech experiments (e.g., on image AR models). The concern that the paper remains a "domain-engineering improvement" rather than a general ML contribution persists.

Theory-Practice Disconnect: The rebuttal admitted that standard MSE was used for "simplicity" despite the theoretical focus on soft value functions. This leaves a conceptual gap in the paper’s primary theoretical anchor.

Computational Efficiency: No specific data on GPU hours or training time was provided to compare VGPO’s online costs with DPO’s offline efficiency.

Metric Choice: The defense of F0V as a "stability benchmark" rather than a "quality proxy" only partially addresses the reviewer's skepticism regarding the real-world utility of the optimized objectives.

**Reviewer Scores:**

Reviewer 1DeF (2->2): This reviewer was fundamentally skeptical about the experimental design and task choice. The authors' explanation that F0V is just an "accessible benchmark" is unlikely to change the reviewer's view on the paper's limited contribution.

Reviewer BGr3 (6 -> 6): This reviewer’s concerns were primarily about missing details and ablations, all of which were addressed in the rebuttal. They would likely maintain their positive stance or slightly increase it.

Reviewer vauG (4 -> 4): The core objection regarding the narrow TTS scope and lack of cross-domain validation was not addressed with new evidence.

Reviewer jkmb (4 -> 4 or 6): The reviewer’s deep technical questions (ADM, DPO details, Normalization) were handled.

---

### Decision · Program_Chairs · 2026-01-26

Reject